# Atorvastatin Treatment Significantly Increased the Concentration of Bone Marrow-Derived Mononuclear Cells and Transcutaneous Oxygen Pressure and Lowered the Pain Scale after Bone Marrow Cells Treatment in Patients with “No-Option” Critical Limb Ischaemia

**DOI:** 10.3390/biomedicines12040922

**Published:** 2024-04-22

**Authors:** Jan Kyselovic, Adriana Adamičková, Andrea Gažová, Simona Valášková, Nikola Chomaničová, Zdenko Červenák, Juraj Madaric

**Affiliations:** 15th Department of Internal Medicine, Faculty of Medicine, Comenius University Bratislava, Špitálska 24, 81372 Bratislava, Slovakia; kyselovic@uniba.sk (J.K.);; 2Department of Pharmacology and Toxicology, University of Veterinary Medicine and Pharmacy, 04181 Košice, Slovakia; 3Institute of Pharmacology and Clinical Pharmacology, Faculty of Medicine, Comenius University Bratislava, Špitálska 24, 81372 Bratislava, Slovakia; 4International Laser Center, Slovak Centre of Scientific and Technical Information, Lamačská cesta 7315/8A, 84104 Bratislava, Slovakia; 5Department of Angiology, Faculty of Medicine, Comenius University and National Institute of Cardiovascular Disease, Pod Krásnou Hôrkou 1, 83101 Bratislava, Slovakia; madaricjuraj@gmail.com

**Keywords:** critical limb ischaemia, bone marrow cell therapy, mesenchymal stem cells, atorvastatin, RAS-acting agents, predictive analysis, spearman correlation

## Abstract

Background: The present study investigated the outcomes and possible predictive factors of autologous bone marrow cells (BMCs) therapy in patients with ”no-option“ critical limb ischaemia (CLI). It was focused on exploring the clinical background and prior statin and renin-angiotensin system (RAS)-acting agents pharmacotherapy related to the therapeutic efficacy of BMCs treatment. Methods: In the present study, we reviewed thirty-three patients (mean age 64.9 ± 10 years; 31 males) with advanced CLI after failed or impossible revascularisation, who were treated with 40 mL of autologous BMCs by local intramuscular application. Patients with limb salvage and wound healing (N = 22) were considered as responders to BMCs therapy, and patients with limb salvage and complete ischemic wound healing (N = 13) were defined as super-responders. Logistic regression models were used to screen and identify the prognostic factors, and a receiver operating characteristics (ROC) curve, a linear regression, and a survival curve were drawn to determine the predictive accuracy, the correlation between the candidate predictors, and the risk of major amputation. Results: Based on the univariate regression analysis, baseline C-reactive protein (CRP) and transcutaneous oxygen pressure (TcPO_2_) values were identified as prognostic factors of the responders, while CRP value, ankle-brachial index (ABI), and bone marrow-derived mononuclear cells (BM-MNCs) concentration were identified as prognostic factors of the super-responders. An area under the ROC curve of 0.768 indicated good discrimination for CRP > 8.1 mg/L before transplantation as a predictive factor for negative clinical response. Linear regression analysis revealed a significant dependence between the levels of baseline CRP and the concentration of BM-MNCs in transplanted bone marrow. Patients taking atorvastatin before BMCs treatment (N = 22) had significantly improved TcPO_2_ and reduced pain scale after BMCs transplant, compared to the non-atorvastatin group. Statin treatment was associated with reduced risk for major amputation. However, the difference was not statistically significant. Statin use was also associated with a significantly higher concentration of BM-MNCs in the transplanted bone marrow compared to patients without statin treatment. Patients treated with RAS-acting agents (N = 20) had significantly reduced pain scale after BMCs transplant, compared to the non-RAS-acting agents group. Similar results, reduced pain scale and improved TcPO_2_, were achieved in patients treated with atorvastatin and RAS-acting agents (N = 17) before BMCs treatment. Results of the Spearman correlation showed a significant positive correlation between CLI regression, responders, and previous therapy before BMCs transplant with RAS-acting agents alone or with atorvastatin. Conclusions: CRP and TcPO_2_ were prognostic factors of the responders, while CRP value, ABI, and BM-MNCs concentration were identified as predictive factors of the super-responders. Atorvastatin treatment was associated with a significantly increased concentration of BM-MNCs in bone marrow concentrate and higher TcPO_2_ and lower pain scale after BMCs treatment in CLI patients. Similarly, reduced pain scales and improved TcPO2 were achieved in patients treated with atorvastatin and RAS-acting agents before BMCs treatment. Positive correlations between responders and previous treatment before BMCs transplant with RAS-acting agents alone or with atorvastatin were significant.

## 1. Introduction

As a terminal stage of peripheral arterial disease (PAD), critical limb ischaemia (CLI), which is characterized by chronic ischemic rest pain, ischemic ulcerations, or gangrene, is associated with a significant risk of amputation of the affected limb [1,2]. Revascularisations by percutaneous transluminal angioplasty or bypass graft surgery are essential rescue methods for wound healing and limb salvage. However, up to 30% of patients with CLI are unsuitable for or have failed previous endovascular or surgical revascularisation treatments [3]. Therefore, in these high-risk patients, limb amputation is the only therapeutic option to relieve pain and stop the spread of wound infection. These patients are called ”no-option“ CLI patients, with an annual mortality of 10% to 40% [4,5].

Several pre-clinical studies of CLI treatment were conducted in recent years, using novel approaches with bone marrow cells (BMCs). The most used stem cells include bone marrow-derived mononuclear cells (BM-MNCs), CD34^+^ bone marrow cells, or bone marrow-derived mesenchymal stem cells. The implanted stem cells can improve blood circulation and increase blood flow to the transplantation site in patients by promoting ischemic angiogenesis and neovascularisation [6,7]. Notably, several clinical trials have demonstrated the positive therapeutic efficacy of stem cell transplantation and validated its safety and feasibility [8,9,10]. At the same time, other studies have shown an insignificant moderate prognosis following stem cell therapy, comparable to the placebo group or conservative treatment [11,12]. The heterogeneity of studied populations and differences in specific features of transplants might cause these controversial observations. Another shortcoming of the cells used in therapy is their poor survival and retention of transplants in vivo, caused by the specific properties of individual cell lines and the hostile microenvironment in the ischemic region. For these reasons, attention has been focused on improving stem cell tolerance to the implantation site, which could increase therapeutic efficacy [13].

In addition, it is necessary to consider previous long-term treatment of CLI patients, which could modify the response to the cell therapy. Recommended therapy in all patients with PAD includes statins, widely used as a cholesterol-lowering agent to prevent cardiovascular disease with a favourable safety profile [14,15]. It has been shown that statins can act through various signalling pathways, particularly PI3K/Akt, thus influencing the fate and properties of transplanted bone marrow cells or affecting the implantation site of stem cells [16]. RAS-acting agents (angiotensin-converting enzyme inhibitors (ACEIs) and angiotensin receptor blockers (ARBs)) should be considered as first-line therapy in patients with PAD and hypertension [14]. A study on 31,245 patients recently concluded that statin therapy positively decreased CRP levels and reduced atherosclerotic risk [17].

While previous research has proposed these theoretical frameworks, their efficacy in real-world clinical settings remains unestablished. Existing therapies for cardiovascular diseases provide a readily available platform for clinically verifying these hypotheses through a simplified clinical design. This study investigates the baseline clinical characteristics and prior medication regimens by ”no-option“ chronic limb ischemia (CLI) patients treated with bone marrow cell therapy. We aim to identify factors associated with treatment response by comparing responders and non-responders. This analysis will contribute to developing and refining selection criteria and prognostic factors for predicting positive clinical outcomes.

## 2. Materials and Methods

### 2.1. Patients

Between May 2016 and November 2018, 33 patients (age 65 ± 10 years, 31 males) with advanced CLI (Rutherford category 5 or 6) after failed or impossible revascularisation were treated with 40 mL of bone marrow nucleated cells via the local intramuscular route. The aetiology of arterial obliteration was atherosclerosis in 30 patients and thromboangiitis obliterans (Buerger disease) in three patients. The patients included in this study met the following criteria: (1) at least 18 years of age with ischemic skin lesions (ulcers or gangrene) with a CLI Rutherford category of 4, 5 or 6 according to the Transatlantic Inter-Society Consensus (TASC) classification (minor or major tissue loss), defined as necrosis or gangrene extending proximal to the metatarsal line or as extensive deep heel gangrene [2]; (2) CLI defined by ankle-brachial index (ABI) ≤ 0.4 or ankle systolic pressure < 50 mmHg, or toe systolic pressure < 30 mmHg, and transcutaneous oxygen pressure (TcPO_2_) < 30 mmHg; (3) no option for endovascular or surgical revascularisation as judged by vascular surgeon and interventionist; (4) failed revascularisation, defined as no change of clinical status with the best standard care four weeks after endovascular or surgical revascularisation. Patients who, at the start of the cell therapy, met any of the following criteria: (1) life expectancy of fewer than six months; (2) presence of malignancy during the last five years; (3) critical coronary artery disease or unstable angina pectoris; (4) patients with end-stage kidney disease and patient on dialysis; (5) bone marrow disease (e.g., severe anaemia, leucopaenia, thrombocytopaenia, myelodysplastic syndrome) were excluded from the study. The local ethical committee of the National Institute of Cardiovascular Diseases, Bratislava, approved the study design. This study was carried out according to the Code of Ethics of the World Medical Association, Declaration of Helsinki (WMA Declaration of Helsinki, 2013).

### 2.2. Bone Marrow Cell Isolation and Administration

On the day of the transplantation, bone marrow was aspirated from both anterior superior iliac crests under analgosedation with propofol. A total volume of 240 mL of bone marrow was harvested using a standard disposable needle for aspiration. Bone Marrow Aspirate Concentrate System (Harvest, Plymouth, MA, USA), which uses gradient density centrifugation and provides 40 mL of bone marrow-rich product for all blood elements within 15 min, was used to process obtained bone marrow aspirate. After the harvesting and centrifugation of stem cells, 40 mL of bone marrow cells (BMCs) were administered by intramuscular methods using deep injections with a 23-G needle into the muscles of the affected limb along the crural arteries, with each injection being approximately 1 mL. The duration of this procedure was 1 h. A bone marrow cell sample was immediately analyzed using a MACS Quant analyzer (Miltenyi Biotec, Bergisch Gladbach, Germany). The BM-MNCs concentration, MSCs concentration, and viability of cells were determined using an MSC Enumerating kit and propidium iodide solution (Miltenyi Biotec, Bergisch Gladbach, Germany), according to manufacturer’s instructions.

### 2.3. Pre-Procedure Assessment and Follow-Up

All patients were examined before the administration of BMCs, as well as 90 days and six months after the delivery, along with a peripheral blood test to determine basal serological parameters (including C-reactive protein, CRP). Resting ABI was measured according to the validated standards [18]. It was performed by measuring the systolic blood pressure from both brachial arteries and dorsalis pedis and posterior tibial arteries after the patient had rested in the supine position for 10 min (normal values, 0.95–1.2). TcPO_2_ of the affected limb was assessed using a TCM400 Mk2 monitor (Radiometer Medical ApS, Copenhagen, Denmark). It was measured at the forefoot in the supine position with an electrode temperature of 44 °C. Wound healing was assessed by two independent physicians and documented by digital photography. A visual analogue scale (VAS) was used to measure pain on a scale graded from 0 to 10.

### 2.4. Endpoints

Primary endpoints were limb salvage and improvement in wound healing without major limb amputation at the six-month of follow-up. Patients who achieved CLI remission were considered responders to BMCs therapy. Patients requiring major limb amputation or patients without signs of wound healing were regarded as non-responders. A group of patients with limb salvage and complete ischemic wound healing were defined as super-responders to BMCs therapy. Patients who underwent major limb amputation were considered as super-non-responders. In addition, patients treated with atorvastatin for a long time (at least six months before BMCs therapy) were included in the atorvastatin group. The other patients were regarded as a non-atorvastatin group. Similarly, patients treated with RAS-acting agents were included in the RAS group, while patients without RAS treatment were considered in the non-RAS-acting agents group. Patients treated with atorvastatin or RAS-acting agents were included in the ATV and RAS group; patients without such therapy were regarded as non-ATV and non-RAS group. The secondary endpoints included changes in TcPO_2_, ABI, pain scale, and Rutherford category after BMCs transplantation. The local ethical committee of the National Institute of Cardiovascular Diseases, Bratislava, approved the study design. All patients in the study were informed about the nature of the study and gave their written informed consent.

### 2.5. Statistical Analysis

The baseline characteristics of the responders and non-responders were compared first. Continuous variables were presented as the mean ± standard deviation (SD). Categorical variables were presented as numbers with percentages, and Fisher’s exact test was used to analyze the significance of the differences. Gaussian distribution of data was tested by the Shapiro-Wilk test. The independent Student *t*-test or Mann-Whitney test was used to analyze the significance of the differences between responders and non-responders. The paired *t*-test or Wilcoxon signed-rank test was used to analyze the longitudinal changes from baseline to 6 months post-transplantation. The candidate prognostic factors of the responders and super-responders were screened through a univariate binary logistic regression analysis. The correlations between the candidate predictors or the variables at different time points were analyzed with linear regression or Spearman nonparametric correlation. Receiver operating characteristics analysis was used to study predictors of clinical response after bone marrow transplantation. The Kaplan–Meier method estimated the survival curve and was compared using the log-rank test. We considered *p* values < 0.05 to be statistically significant. All statistical analyses were performed using Microsoft Office Excel (2007), XLSTAT statistical and data analysis solution (Addinsoft 2021, New York, NY, USA) or GraphPad Prism version 9.0 (GraphPad Software, San Diego, CA, USA).

## 3. Results

### 3.1. Baseline Information and Endpoints

The mean age of 33 patients included and analyzed in the study was 64.9 ± 10 years, and the male ratio was 94%. There were 32 lower limbs and one upper limb involved in the study. The baseline Rutherford class was 5 in 32 patients (97%) and 6 in one patient (3%) (Table 1). The six-month major amputation-free survival rate was 81.8% (27/33), wherein six patients underwent major amputations due to CLI progression. The combined primary endpoint of limb salvage and wound healing was met in 22 of the 33 patients (66.7%) at the six-month follow-up and were regarded as responders to the BMCs therapy. A total of 13 patients (39.4%) achieved limb salvage and complete wound healing at six months and were regarded as super-responders to the BM-MNCs transplantation (Figure 1).

### 3.2. Characteristics of Responders and Non-Responders

The baseline information of patients before the BMCs therapy is presented in Table 1. The mean age of the responders was higher than that of the non-responders; the difference was significant (67.4 ± 9.3 vs. 59.8 ± 10.7, *p* = 0.014). No significant variations regarding limb ischaemia risk factors were observed between the responders and non-responders. When the baseline blood parameters were evaluated, the responders had significantly lower CRP levels than the patients in the non-responders category (*p* = 0.013).

Regarding the treated limbs, the responders were characterized by fewer patients with baseline TcPO_2_ < 10 mmHg than the non-responders (*p* = 0.278). Based on the available complete longitudinal data (N = 28), the mean TcPO_2_ of the responders increased from 12.5 ± 14.0 mmHg at baseline to 22.7 ± 15.2 mmHg at six months post-transplantation (*p* = 0.008). At the same time, that of the non-responders did not improve significantly, from 2.4 ± 0.63 mmHg at baseline to 3.18 ± 8.4 mmHg at six months post-transplantation (*p* = 0.573) (Table 2, Figure 2A). TcPO_2_ at baseline correlated significantly with that at six months (R^2^ = 0.371, *p* = 0.001) (Figure 2B).

No differences were observed between the responders and the non-responders in baseline ABI (*p* = 0.183), and there was no significant improvement during the six-month follow-up. Despite this, a significant ABI decrease was observed in the non-responder group (*p* = 0.043). Moreover, we did not find a significant variation in the pain scale between groups at the baseline (*p* = 0.366). Still, a significant pain reduction was observed in the responder group during the six-month follow-up (*p* < 0.001). We did not see any significant changes in the non-responder group regarding the pain scale (*p* = 0.445). In addition, no difference was revealed between groups in terms of the Rutherford class at the baseline (*p* = 0.208). Still, significant improvement was observed at the six-month follow-up in the responder group (*p* = 0.008), with no difference in the non-responder group (*p* = 0.059) (Table 2).

### 3.3. Characteristics of Super-Responders and Super-Non-Responders

Table 3 presents baseline information of patients before BMCs therapy. The mean age of super-responders tended to be lower than that of the super-non-responders, though the difference was not significant (64.8 ± 10.4 vs. 66.8 ± 4.0, *p* = 0.690). Comparing blood parameters at baseline demonstrated lower CRP levels in the super-responder group than patients in the super non-responder group (*p* < 0.001).

The super-responders were characterized by higher baseline TcPO_2_ (14.17 ± 14.69 mmHg) and ABI (0.57 ± 0.37) than the super-non-responders (TcPO_2_ 3.0 ± 2.65 mmHg, *p* = 0.536, ABI 0.18 ± 0.36, *p* = 0.101); however, these differences were not statistically significant (Table 3). Based on the available complete longitudinal data, the mean TcPO_2_ of the super-responders increased from 14.17 ± 14.69 mmHg at baseline to 25.92 ± 14.92 mmHg at six months post-transplantation (*p* = 0.012). In contrast, that of the super-non-responders did not improve because of the deteriorating health and amputation of the affected limb. There was no significant improvement in the case of ABI of the super-responders (0.57 ± 0.37 at baseline, 0.57 ± 0.29 at six months, *p* = 0.983), while significant improvement of pain scale was demonstrated in the super-responders group (5.61 ± 1.71 at baseline, 1.46 ± 1.2 at six months post-transplantation, *p* < 0.0001), compared to the worsened pain scale in the super-non-responder group (6.33 ± 0.58 at baseline, 9.0 at six months, *p* = 0.102). Regarding the Rutherford scale, the super-responder group had significantly improved from 5.0 at baseline to 3.92 ± 1.04 at six months post-transplantation (*p* = 0.008). In comparison, the Rutherford scale decreased from 4.83 ± 0.75 at baseline to 6.0 (*p* = 0.038) at six months post-transplantation.

### 3.4. Characteristics of Transplanted Bone Marrow Cells

According to the characteristics of BMCs transplant, several variables differed between groups (Table 4). The concentration of BM-MNCs appeared to be higher in the responders (3.53 ± 1.5 × 10^9^) than in the non-responders (2.86 ± 1.13 × 10^9^), although the difference was not statistically significant (*p* = 0.213). However, in the case of subgroup analysis of super-responders and super-non-responders, we found a significantly higher number of BM-MNCs in the super-responders compared to super-non-responders (3.68 ± 1.51 × 10^9^ vs. 2.26 ± 0.74 × 10^9^, *p* = 0.049).

### 3.5. Prognostic Factors and Predictors of BMCs Treatment Outcomes

According to the results of the univariate logistic regression, two variables were screened out for responders: CRP [OR 0.955, 95% CI 0.901–1.012, *p* = 0.044] and TcPO_2_ [OR 1.149, 95% CI 0.949–1.392, *p* = 0.021] (Table 5). The results of univariate logistic regression for super-responders showed that CRP [OR 0.544, 95% CI 0.221–1.341, *p* < 0.0001], ABI [OR 56.140, 95% CI 0.531–5932.9, *p* = 0.035] and BM-MNCs concentration [OR 1.0, 95% CI 0.999–1.0, *p* = 0.035] were screened out for super-responders (Table 5).

The receiver operating characteristics (ROC) analysis (Figure 3) was performed based on the univariate logistic regression results for the responder. An ROC curve was generated to address the sensitivity and specificity of CRP. The area under the ROC curve for CRP was 0.768 (95% CI 0.572–0.96, *p* = 0.014), indicating good discrimination for non-responders. Analysis showed that a cut-off limit of baseline CRP > 8.1 mg/L before transplantation was predictive for negative clinical response, with 81% specificity and 82% sensitivity (Figure 3A). Analysis of linear regression showed a significant dependence between the levels of baseline CRP and the concentration of BM-MNCs in transplanted bone marrow (R^2^ = 0.162, *p* = 0.03) (Figure 3B). 

### 3.6. Effect of Atorvastatin Therapy before BMCs Treatment on the Outcomes of Stem Cell Treatment

The patients’ baseline characteristics before BMCs therapy are presented in Table 6. No significant differences were observed between the atorvastatin and non-atorvastatin groups regarding the basic characteristics, previous blood examination, or parameters of limb ischaemia at baseline (Table 6).

### 3.7. Parameters of Limb Ischaemia after BMCs Delivery in Atorvastatin or Non-Atorvastatin Group

Based on the functional outcomes after BMCs treatment, significant improvement of TcPO_2_ was observed in the atorvastatin group (*p* = 0.015), compared to nonsignificant improvement in the non-atorvastatin group (*p* = 0.611) (Table 7). Reduction in pain scoring was statistically significant in the atorvastatin group (*p* = 0.004), while no difference was observed in the non-atorvastatin group (*p* = 0.202). Amputation-free survival was achieved in 18 patients (82%) in the ATV group, while in the non-ATV group was achieved in 8 patients (73%, *p* = 0.661). The risk of major amputation was decreased in those prescribed statins (Hazard ratio (HR) 0.44, 95% CI 0.08–2.5, *p* = 0.36) but did not differ significantly (Figure 4C).

### 3.8. Characteristics of Transplanted Bone Marrow Cells in ATV and Non-ATV Group

Table 8 compares the characteristics of BMCs transplants administered to the affected limb. The concentration of BM-MNCs was significantly higher in the atorvastatin group (3.64 ± 1.53 × 10^9^) than in the non-atorvastatin group (2.58 ± 0.73 × 10^9^, *p* = 0.038) (Table 8). No other statistically significant differences were observed between groups in terms of cell viability and count of MSCs in the transplant.

### 3.9. Effect of RAS-Acting Agents Therapy Prior to BMCs Treatment on the Outcomes of Stem Cell Treatment

Table 9 presents the baseline characteristics of patients before BMCs therapy. Patients with RAS-acting agents treatment were characterized by higher age (*p* = 0.006) and higher rate of arterial hypertension (*p* = 0.025).

### 3.10. Parameters of Limb Ischaemia after BMCs Delivery in RAS or Non-RAS Group

Based on the functional outcomes after BMCs treatment, significant improvement in pain scale was observed in the RAS group (*p* = 0.005), compared to nonsignificant improvement in the non-RAS group (*p* = 0.139). We have not seen any significant changes in the RAS or non-RAS group comparing Rutherford class, TcPO_2_, or ABI during the six-month follow-up (Table 10).

In the RAS group, 16 patients (80%) achieved major amputation-free survival six months post-transplantation, while in the non-RAS group amputation-free survival was achieved in 10 patients (77%, *p* = 1.0). The risk of major amputation was decreased in those prescribed RAS-acting agents (HR 0.438, 95% CI 0.08–2.4, *p* = 0.34) but did not differ significantly (Figure 5).

### 3.11. Characteristics of Transplanted Bone Marrow Cells in RAS or Non-RAS Group

No statistical differences were observed between the RAS and non-RAS groups in terms of characteristics of BMCs transplants administered to the affected limb (Table 11).

### 3.12. Effect of Atorvastatin and RAS-Acting Agents Therapy Prior to BMCs Treatment on the Outcomes of Stem Cell Treatment

Of the 33 patients, 17 were treated with atorvastatin together with RAS-acting agents before stem cell treatment, while eight patients were without atorvastatin and RAS-acting agents treatment. The baseline characteristics of these groups were compared first, with only a significant difference in the mean age of patients (*p* = 0.026) (Table 12).

### 3.13. Parameters of Limb Ischaemia after BMCs Delivery in ATV and RAS Group or Non-ATV and Non-RAS Group

According to the results of BMCs treatment after six months, several variables were screened out in the ATV and RAS group (Table 13). Based on the available complete longitudinal data, the mean TcPO_2_ of the RAS and ATV group increased from 11.6 ± 13.5 mmHg at baseline to 16.5 ± 15.0 mmHg at six months post-transplantation (*p =* 0.033). In contrast, that of the non-ATV and non-RAS groups did not improve significantly (*p* = 0.344). Regarding the results of improvement in the pain scale, the ATV and RAS group was associated with significant improvement of the pain scale from 5.80 ± 1.42 at baseline to 3.50 ± 3.43 at six months (*p =* 0.009), while assessment of the pain scale did not change significantly in the non-ATV and non-RAS group (*p =* 0.279). Amputation-free survival was achieved in 13 patients (76%) in the ATV and RAS group, while in the non-ATV and non-RAS group it was achieved in 5 patients (62.5%, *p* = 0.344). The risk of major amputation was decreased in those prescribed atorvastatin and RAS-acting agents (HR 0.39, 95% CI 0.07–2.3, *p* = 0.36) but did not differ significantly (Figure 6).

### 3.14. Characteristics of Transplanted Bone Marrow Cells in ATV and RAS Group or Non-ATV and Non-RAS Group 

Regarding the transplants, the transplanted BM-MNCs count appeared to be higher in the ATV and RAS group (3.56 ± 1.60 × 10^9^) than in the non-ATV and non-RAS group (2.60 ± 0.81 × 10^9^). However, the difference was not statistically significant (*p* = 0.064) (Table 14).

### 3.15. Results of Spearman Correlation Analysis for the Investigated Treatment Option

The Spearman correlation between investigated treatment options before BMCs transplantation and baseline characteristics is shown in Table 15. Patients with CLI regression, patients considered as responders, showed a significant positive correlation with age (R = 0.430, *p* = 0.012) and RAS treatment (R = 0.351, *p* = 0.045). Relevantly, there was a negative correlation with CRP (R = −0.442, *p* = 0.011). Statin treatment before BMCs administration correlated with RAS treatment (R = 0.482, *p* = 0.005). RAS-acting agents treatment was highly correlated with age of patients (R = 0.566, *p* = 0.0006). Treatment with both investigated pharmacotherapies, statin and RAS treatment, showed a significant positive correlation with CLI regression, responder status (R = 0.618, *p* = 0.001), and age of patients (R = 0.454, *p* = 0.023).

## 4. Discussion

### 4.1. Summary of the Results

This study investigated the outcomes and possible predictive factors of autologous bone marrow cell transplantation in treating ”no-option“ CLI patients. It was focused on exploring clinical background and prior statin pharmacotherapy related to the therapeutic efficacy of BMCs treatment. Our results revealed that CRP and TcPO_2_ values positively correlated with the probability of being a responder. In contrast, CRP value, ABI, and BM-MNCs concentration were identified as prognostic factors of the super-responders. Based on the results of univariate logistic regression for responders, ROC analysis of CRP revealed that a cut-off limit of CRP > 8.1 mg/L before transplantation was predictive for negative clinical response after BMCs transplantation. Linear regression analysis demonstrated a significant dependence between the levels of baseline CRP and the concentration of BM-MNCs in transplanted bone marrow.

Atorvastatin treatment prior to BMCs transplant revealed significant improvement of TcpO_2_ and reduction of pain scale after BMCs transplant compared to nonsignificant changes in the non-atorvastatin group. Our results showed that atorvastatin use was associated with decreased risk for major amputation. However, the difference was not statistically significant. Furthermore, our findings showed a significantly higher concentration of BM-MNCs in the transplanted bone marrow cells of patients in the atorvastatin group than in the non-atorvastatin group, which might contribute to the beneficial effect of cell therapy.

Patients treated with RAS-acting agents before BMCs transplant had significantly reduced pain scale after six months, compared to the non-RAS group. Similar results, reduced pain scale as well as improved TcPO_2_, were achieved in a group of patients treated with atorvastatin and RAS-acting agents before BMCs treatment. Results of Spearman correlation showed a significant positive correlation between CLI regression, patients considered as responders, and previous treatment before BMCs transplant with RAS-acting agents alone or together with atorvastatin.

### 4.2. Prognostic Factors of the Therapeutic Responses to Autologous BMCs Treatment

As the terminal stage of peripheral arterial disease, CLI is characterized by an extremely high risk of amputation and other vascular issues, causing severe morbidity and mortality in affected patients. Therapeutic angiogenesis with cell-based therapies aims to increase blood flow to ischemic regions. One of the most encouraging cells used as an alternative for the surgical treatment of CLI is mesenchymal stem cells appearing in the bone marrow-derived mononuclear stem cells population [19]. Although stem cell therapies are studied clinically for their benefit in the treatment of cardiac repair or patients with CLI, their effects are still controversial and considered experimental [13,20,21]. The published data show that treatment with bone marrow stem cells is associated with limb salvage, increased TcPO_2_, ABI, or blood flow perfusion [22]. The results of our study present the beneficial effects of autologous BMCs transplantation in TcPO_2_, pain scoring, and Rutherford category, but not ABI. ABI did not change after six months, similar to the results of other authors [23,24], but it was inconsistent with the report of Gupta et al. 2013 and Benoit et al. 2013 [25,26]. The prevalence of amputation-free survival at 82% was similar to the results of Zafarhandi et al. 2010, who also reported the effects of cell therapy after six months [27].

TcPO_2_ represents the tension of oxygen disseminated from subcutaneous microcirculation and can reflect the distal peripheral perfusion [28]. TcPO_2_ measurement is a metabolic test and is considered a helpful predictor for chronic ischemic ulcer healing, with a threshold value of 20–40 mmHg [29]. On the contrary, the unfavourable value for spontaneous healing is less than 10 mmHg [30,31]. In our study, the evaluation of the baseline characteristics showed that responders had higher baseline TcPO_2_ than non-responders, but the difference was not statistically significant. The univariate regression analysis showed that the baseline TcPO_2_ value was a prognostic factor for being a responder to BMCs treatment.

Moreover, according to the longitudinal data, we found significant improvement of TcPO_2_ at six months post-transplantation in the responder group, whereas little improvement was found in the non-responders group. Moreover, the TcPO_2_ at baseline correlated significantly with values at six months post-transplantation. Based on these data, we assume that baseline TcPO_2_ represents an important criterion for the patients undergoing transplantation of bone marrow stem cells, consistent with several previous studies [29,32,33]. These results suggest that the restoration of peripheral perfusion after BMCs therapy is characterized by a favourable baseline condition of local microcirculation, which might explain the predictive role of baseline TcPO_2_.

Inflammatory markers such as C-reactive protein (CRP), fibrinogen, and serum amyloid A are associated with an increased risk of cardiovascular events. They are considered critical risk factors for the development and progression of PAD [34,35]. Our study demonstrates that responders to BMCs therapy are characterized by significantly lower baseline CRP levels than non-responders. The univariate regression analysis showed that the baseline CRP value was a prognostic factor for being a responder to BMCs treatment. CRP belongs to acute-phase reactant protein and is produced during inflammation, which negatively modulates the local inflammatory reaction in the transplantation site and, thus, the entire process of therapeutic angiogenesis after BM-MNCs transplantation [36]. Moreover, CRP influences the systemic inflammatory reaction and can negatively regulate bone marrow cells’ characteristics. Indeed, we found that depressed numbers of BM-MNCs in the bone marrow transplant were associated with elevated levels of CRP in patients with critical limb ischaemia. We found that these two characteristics are significantly dependent upon each other. The receiver operating characteristics analysis showed a cut-off limit of baseline CRP > 8.1 mg/L before transplantation was predictive for negative clinical response after BMCs transplantation.

In addition to the patient’s clinical background, the characteristic of the bone marrow transplant plays a critical role in the healing prognosis. An important predictor of the therapeutic response is the number of administered nucleated cells, bone marrow-derived mononuclear cells, which strongly relate to clinical benefit in the PROVASA trial and are considered an independent predictor of improved ulcer healing [12]. On the other hand, in the study of Yusoff et al. 2020, the authors found that the difference in the number of BM-MNCs derived from bone marrow did not alter the major amputation-free survival rate or mortality rate in atherosclerotic PAD patients with CLI [37]. Our results demonstrate a significant difference in the concentration of BM-MNCs, however, in the subgroup analysis of super-responders, patients with limb salvage and complete ischemic wound healing, compared to super-non-responders, patients with major limb amputation. Specifically, a significantly higher number of BM-MNCs was present in the BMCs transplant of the super-responder group compared to the super-non-responder group.

Moreover, the univariate regression analysis showed that the concentration of BM-MNCs was a prognostic factor for being a super-responder to BMCs treatment. Accordingly, we believe that the concentration of BM-MNCs in transplant might be feasible for NO-CLI patients. A longer-term follow-up study will be needed to compare the outcomes and advantages of transplantation types.

### 4.3. Subgroup Analysis of Treatment Approach before BMCs Transplant

Many studies demonstrated the poor survival and retention of transplanted bone marrow cells in vivo, caused by stem cells’ properties, the extremely hostile microenvironment of implantation, or a combination of both [13]. For these reasons, the effort has been focused on improving the tolerance of transplanted stem cells to the microenvironment. This would lead to developing a clinical approach with enhanced stem cell survival and angiogenesis in the implantation site. Recent pre-clinical and clinical studies describe the use of statin drugs to augment the function of MSCs or endothelial progenitor cells (EPC) for regenerative cell-based therapy. Promising results have been reported in treating AMI in animal models but have yet to be reported in human studies. Pre-clinical and clinical studies reported a beneficial effect of statin therapy that improves MSCs or EPC function and their numbers. However, statin use in enhancing cell-based vascular repair warrants further study [38,39,40]. Systematic review and meta-analysis of nineteen studies with 26,985 patients with CLI showed that statin use was associated with decreased risk of major amputation, mortality, and more remarkable amputation-free survival and overall patency rates [16].

In this study, we examined the efficacy of atorvastatin treatment before BMCs transplant on outcomes of cell-based therapy. Our data confirmed a significant improvement of TcPO_2_ and a reduction of pain scoring in the atorvastatin group, compared to a nonsignificant improvement in the non-atorvastatin group. Our results showed that statin use was associated with a decreased risk of major amputation. However, the difference was not statistically significant. Furthermore, our findings showed a significantly higher concentration of BM-MNCs in transplanted bone marrow cells in patients with prior atorvastatin treatment compared to the non-atorvastatin group. We suggest that these beneficial effects of atorvastatin contribute to the valuable effect of cell therapy in CLI patients and are related to the clinical benefit of stem cell treatment.

Moreover, we focused on the effects of RAS-acting agents treatment before BMCs transplant on the outcomes of cell-based therapy. RAS-acting agents (angiotensin-converting enzyme inhibitors (ACEIs) and angiotensin receptor blockers (ARBs)) should be considered as first-line therapy in patients with PAD and hypertension [14]. Bodewes et al. 2018 revealed the long-term mortality benefit of RAS inhibitors in chronic limb-threatening ischaemia patients, whereas limb events were unaffected [41]. Our study has demonstrated that patients treated with RAS-acting agents before BMCs transplant had significantly reduced pain scale after six months, compared to the non-RAS group. Similar results, reduced pain scale and improved TcPO_2,_ were achieved in patients treated with atorvastatin and RAS-acting agents before BMCs treatment.

Results of the Spearman correlation showed a significant positive correlation between CLI regression, responders, and previous treatment before BMCs transplant with RAS-acting agents alone or with atorvastatin.

## 5. Limitations

There were several limitations to this study. First, no parallel placebo control group is present due to ethical concerns and one single centre. Additionally, the sample size was insufficient to evaluate more specific variables. Moreover, our study focused on short-term responders (six months) to BMCs therapy rather than long-term remission from CLI, which is also the target of therapeutic angiogenesis. Therefore, a prolonged cohort follow-up with larger numbers of patients is needed in the future.

## 6. Conclusions

Our data suggest the potential benefits of carefully managing patients with CLI before BMCs transplantation, as this approach appears to enhance the effectiveness of cell therapy.

CRP and TcPO_2_ values were positively associated with a positive response to BMCs transplantation. Additionally, our analyzed factors (CRP value, ABI, and BM-MNCs concentration) may help predict responders and super-responders to cell therapy. A baseline CRP level above 8.1 mg/L could indicate a reduced positive clinical response after BMCs transplantation. In line with recommended guidelines for treating CLI patients, long-term atorvastatin use appears to improve outcomes after BMCs therapy, potentially through increased BM-MNCs in the transplant. The long-term treatment of CLI patients with RAS-acting agents significantly reduced pain scale after BMCs transplant. The combination of atorvastatin and RAS-acting agents shows promise for reducing pain and improving TcPO2 values in CLI patients after BMCs therapy.

While these findings are encouraging, further research is needed to confirm them in larger studies, to explore the exact mechanisms by which statins and RAS-acting agents might improve BMCs therapy outcomes, and to determine the optimal timing and combination of these therapies with BMCs transplantation.

## Figures and Tables

**Figure 1 biomedicines-12-00922-f001:**
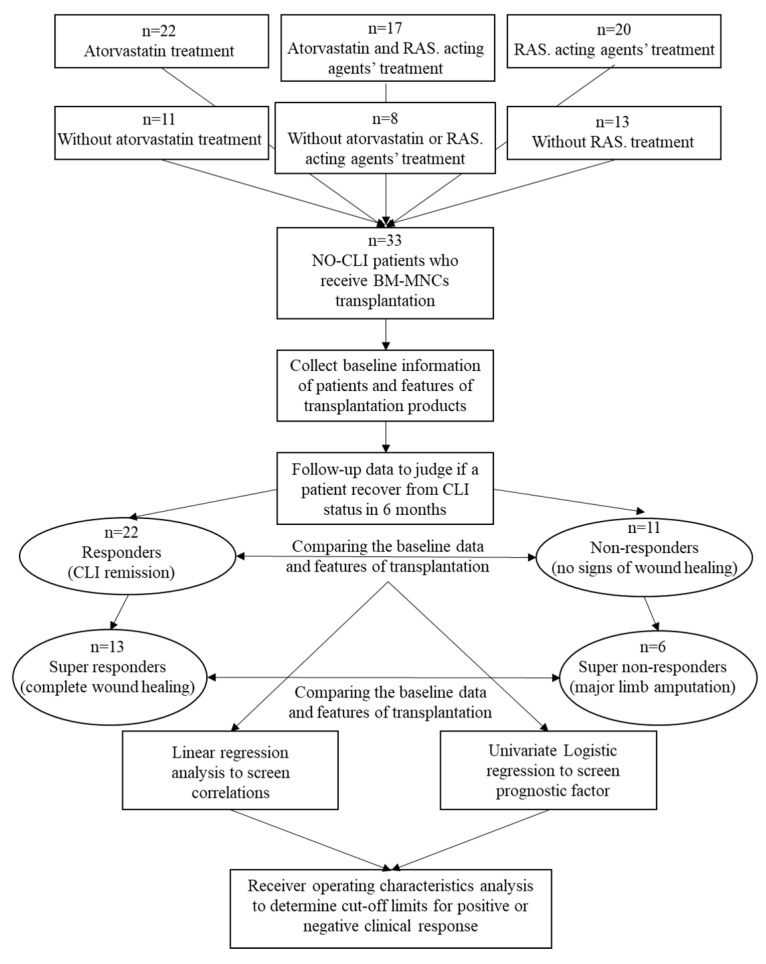
Study flow diagram. NO-CLI, “no-option” critical limb ischaemia.

**Figure 2 biomedicines-12-00922-f002:**
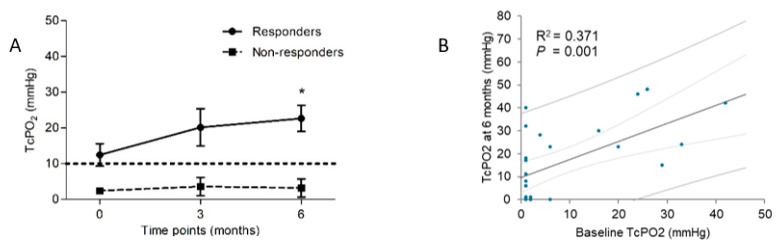
Comparison of longitudinal TcPO_2_ changes (mean with SEM) in the ischemic limbs of the responders and non-responders (**A**) and linear regression between TcPO_2_ at baseline and at six months post-transplantation, depicted with a solid fitting line, dotted 95% confidence interval bars and lined 95% prediction intervals (**B**). * The difference between six months and baseline in responders was statistically significant (*p* = 0.008). TcPO_2_, transcutaneous oxygen pressure.

**Figure 3 biomedicines-12-00922-f003:**
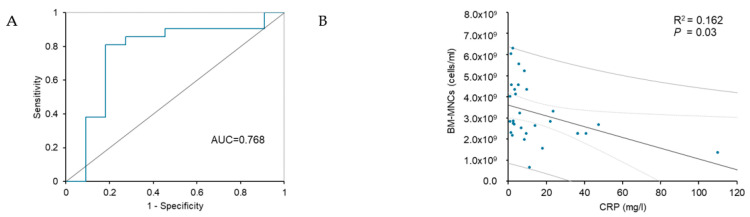
Receiver operating characteristics of CRP levels for predicting the BMCs therapeutic response. Area under the receiver operating characteristic (ROC) curve: CRP = 0.768 (CI 0.572–0.96, *p* = 0.014) (**A**) and linear regression between CRP and BM-MNCs, with lined 95% confidence interval bars and lined 95% prediction intervals (**B**). CRP, C-reaction protein; BMCs, bone marrow cells; BM-MNCs, bone marrow-derived mononuclear cells.

**Figure 4 biomedicines-12-00922-f004:**
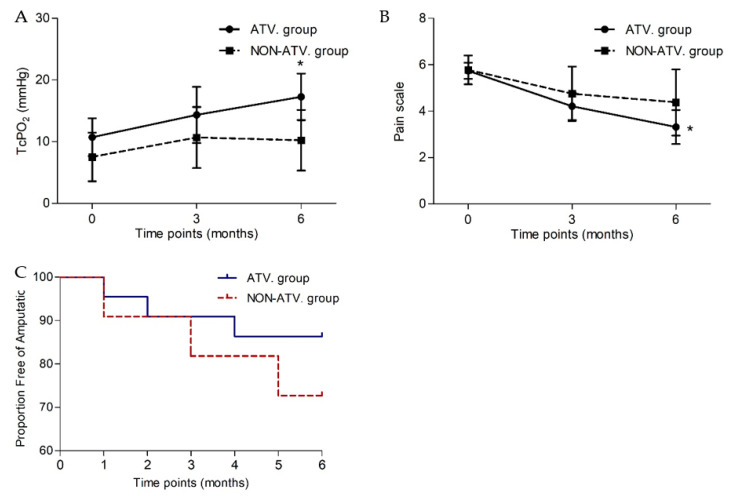
Comparison of longitudinal TcPO_2_ changes (mean with SEM) in the ischemic limbs of patients treated with or without atorvastatin prior to BMCs treatment (**A**); comparison of longitudinal pain scale changes (mean with SEM) in the ischemic limbs of patients treated with or without atorvastatin prior to BMCs treatment (**B**); Kaplan-Meier curve to six months post-transplantation showing the proportion free of amputation (**C**). TcPO_2_, transcutaneous oxygen pressure. * The difference between six months and baseline in responders was statistically significant.

**Figure 5 biomedicines-12-00922-f005:**
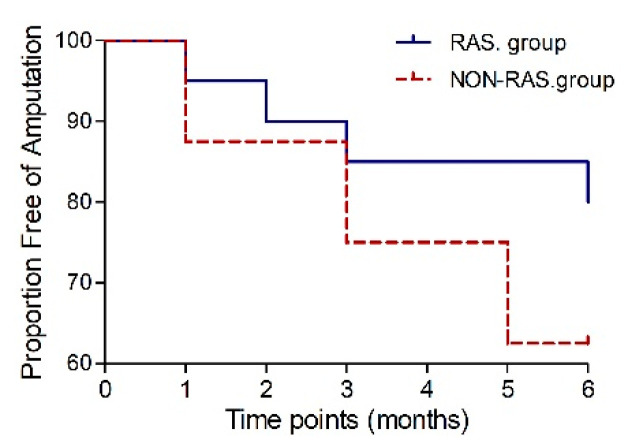
Kaplan-Meier curve to six months post-transplantation showing the proportion free of amputation.

**Figure 6 biomedicines-12-00922-f006:**
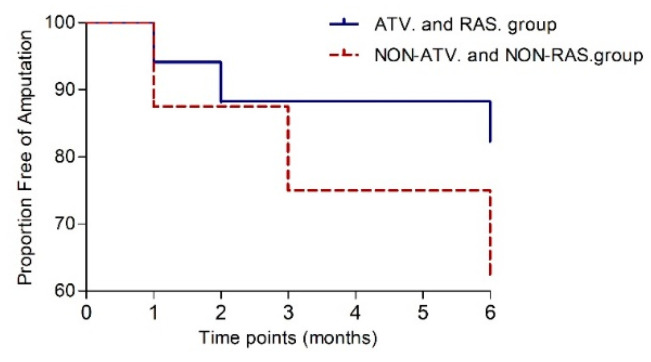
Kaplan-Meier curve to six months post-transplantation showing the proportion free of amputation.

**Table 1 biomedicines-12-00922-t001:** Characteristics of patients before stem cell treatment.

	All Patients (N = 33)	Responders (N = 22)	Non-Responders (N = 11)	*p* Value
Age (years)	64.9 ± 10	67.4 ± 9.3	59.8 ± 10.7	0.014 *
Sex (males)	31 (94%)	21 (95%)	10 (91%)	1.000
Rutherford class (1–6)	5.0 ± 0.30	5.0	4.9 ± 0.54	0.208
Body mass index (kg/m^2^)	27 ± 4.1	27.2 ± 3.8	27.6 ± 4.9	0.756
	Risk factors of limb ischaemia			
Diabetes mellitus	11 (33%)	7 (32%)	4 (36%)	1.000
Arterial hypertension	27 (82%)	18 (82%)	9 (82%)	1.000
Hyperlipidaemia	23 (70%)	15 (68%)	8 (73%)	1.000
Smoker	10 (30%)	6 (27%)	4 (36%)	0.696
	Blood examination			
CRP (mg/L)	13.3 ± 21.3	8.1 ± 12.3	23.3 ± 30.5	0.013 *
Creatinine (µmol/L)	95.9 ± 34.6	95.4 ± 35.7	96.7 ± 34.2	0.946
	Treatment history			
Statins	22 (67%)	16 (73%) (NNT 5.5)	6 (55%)	0.255
Antiplatelet drugs	24 (73%)	17 (77%)	7 (64%)	0.438
Aspirin	16 (48%)	11 (50%)	5 (45%)	0.4074
Clopidogrel	17 (52%)	10 (45%)	7 (64%)	0.1711
RAS-acting agents	20 (61%)	16 (73%) (NNT 2.75)	4 (36%)	0.065
Statins and RAS-acting agents	17 (52%)	13 (59%) (NNT4.4)	4 (36%)	0.282
Naftidrofuryl	28 (85%)	18 (82%)	10 (91%)	0.643
Post-MI	31 (94%)	22 (100%)	9 (82%)	0.104
	Parameters of limb ischaemia			
TcPO_2_ < 10 mmHg	19 (58%)	11 (50%)	8 (73%)	0.278
ABI	0.51 ± 0.38	0.57 ± 0.35	0.37 ± 0.43	0.183
Pain scale (0–10)	5.75 ± 1.6	5.62 ± 1.75	6.14 ± 1.07	0.366

Where appropriate, data are presented as mean ± standard deviation or numbers and percentages. CRP, C-reaction protein; RAS, renin-angiotensin system; MI, myocardial infarct; TcPO_2_, transcutaneous oxygen pressure; ABI, ankle-brachial index; NNT, number needed to treat. Statistical analysis was performed by the Mann-Whitney test and by Student’s *t*-test (depending on data distribution determined by the Shapiro-Wilk test) or Fisher’s exact test; *p*-value = responders vs. non-responders. * The difference between six months and baseline in responders was statistically significant.

**Table 2 biomedicines-12-00922-t002:** Outcomes of BMCs treatment after six months in responders and non-responders.

**Parameters of Limb Ischaemia**	**Responders**			**Non-Responders**		
	Baseline (N = 22)	Six Months (N = 17)	*p* Value	Baseline (N = 11)	Six Months (N = 11)	*p* Value
Rutherford class (1–6)	5.0 ± 0.30	4.2 ± 1.0	0.008 *	4.9 ± 0.54	5.55 ± 0.52	0.059
TcPO_2_ (mmHg)	12.5 ± 14.0	22.7 ± 15.2	0.008 *	2.4 ± 1.77	3.18 ± 8.4	0.573
ABI	0.57 ± 0.35	0.60 ± 0.34	0.480	0.37 ± 0.43	0.029 ± 0.08	0.043 *
Pain scale (0–10)	5.62 ± 1.75	1.78 ± 1.22	<0.001 *	6.14 ± 1.07	6.86 ± 3.76	0.445

Data are presented as mean ± standard deviation. ABI, ankle-brachial index; TcPO_2_, transcutaneous oxygen pressure. Statistical analysis was performed by the Wilcoxon matched-pair test and Student’s *t*-test (depending on data distribution determined by the Shapiro-Wilk test). *p* = six months vs. baseline.* The difference between six months and baseline in responders was statistically significant.

**Table 3 biomedicines-12-00922-t003:** Characteristics of super-responders and super-non-responders before stem cell treatment.

	All Patients (N = 33)	Super-Responders (N = 13)	Super-Non-Responders (N = 6)	*p* Value
Age (years) (mean ± SD)	64.9 ± 10	64.8 ± 10.4	66.8 ± 4.0	0.690
Sex (males)	31 (94%)	12 (92%)	6 (100%)	1.000
Rutherford class (1–6)	5.0 ± 0.30	5.0	4.83 ± 0.75	0.175
Body mass index (kg/m^2^)	27 ± 4.1	27.1 ± 4.04	25.6 ± 3.76	0.450
	Risk factors of limb ischaemia			
Diabetes mellitus (N, %)	11 (33%)	5 (38%)	3 (50%)	1.000
Arterial hypertension	27 (82%)	11 (85%)	6 (100%)	1.000
Hyperlipidaemia	23 (70%)	10 (77%)	5 (83%)	1.000
Smoker	10 (30%)	3 (23%)	2 (33%)	1.000
	Blood examination			
CRP (mg/L)	13.3 ± 21.3	5.2 ± 3.45	36.6 ± 37.01	<0.001 *
Creatinine (µmol/L)	95.9 ± 34.6	98.7 ± 41.02	102.5 ± 39.78	0.914
	Treatment history			
Statins (N, %)	22 (67%)	11 (85%)	3 (50%)	0.262
Antiplatelet drugs	24 (72%)	12 (92%)	3 (50%)	0.071
Aspirin	16 (48%)	8 (62%)	2 (33%)	0.1463
Clopidogrel	17 (52%)	8 (62%)	3 (50%)	0.3361
RAS-acting agents	20 (61%)	9 (69%)	3 (50%)	0.617
Statins and RAS-acting agents	17 (52%)	9 (69%)	2 (33%)	0.319
Naftidrofuryl	28 (85%)	12 (92%)	6 (100%)	1.000
Post-MI	31 (94%)	12 (92%)	5 (83%)	1.000
	Parameters of limb ischaemia			
TcPO_2_ < 10 mmHg	19 (58%)	6 (46%)	4 (67%)	0.629
ABI	0.51 ± 0.38	0.57 ± 0.37	0.18 ±0.36	0.101
Pain scale (0–10)	5.75 ± 1.6	5.61 ± 1.71	6.33 ± 0.58	0.907

Where appropriate, data are presented as mean ± standard deviation or numbers and percentages. CRP, C-reaction protein; RAS, renin-angiotensin system; MI, myocardial infarct; TcPO_2_, transcutaneous oxygen pressure; ABI, ankle-brachial index. Statistical analysis was performed by the Mann-Whitney test and by Student’s *t*-test (depending on data distribution determined by the Shapiro-Wilk test) or Fisher’s exact test; *p*-value = responders vs. non-responders. * The difference between six months and baseline in responders was statistically significant.

**Table 4 biomedicines-12-00922-t004:** Characteristics of transplanted BMCs.

Group Analysis	Responders (N = 22)	Non-Responders (N = 11)	*p* Value
BM-MNCs (109 cells/mL)	3.53 ± 1.5	2.86 ± 1.13	0.213
Viability of BM-MNCs (%)	99.70 ± 0.3	99.27 ± 0.86	0.464
MSCs (104 cells/mL)	0.79 ± 0.61	1.46 ± 1.56	0.123
Subgroup analysis	super-responders (limb salvage and complete ischemic wound healing, N = 13)	super-non-responders (major limb amputation, N = 6)	*p* value
BM-MNCs (109 cells/mL)	3.68 ± 1.51	2.26 ± 0.74	0.049 *
Viability of BM-MNCs (%)	99.71 ± 0.32	99.05 ± 0.94	0.263
MSCs (104 cells/mL)	0.83 ± 0.52	1.68 ± 2.14	0.639

Data are presented as mean ± standard deviation. BMCs, bone marrow cells; BM-MNCs, bone marrow mononuclear cells; MSCs, mesenchymal stem cells. Statistical analysis was performed by the Mann-Whitney test and Student’s *t*-test (depending on data distribution determined by the Shapiro-Wilk test). *p* = responders vs. non-responders; super-responders vs. super-non-responders. * The difference between six months and baseline in responders was statistically significant.

**Table 5 biomedicines-12-00922-t005:** Univariate logistic regression analysis of prognostic factors.

Candidate Variable	Responder		Super-Responder	
	OR (95% CI)	*p* Value	OR (95% CI)	*p* Value
Age ≥ 50 years	1.097 (0.993–1.212)	0.056	0.972 (0.844–1.095)	0.689
Rutherford class	2.803 (0.221–35.552)	0.411	3.026 (0.222–41.289)	0.389
Body mass index (kg/m^2^)	0.977 (0.819–1.166)	0.800	1.118 (0.847–1.476)	0.412
Smoker	0.808 (0.169–3.858)	0.653	0.500 (0.050–4.978)	0.712
CRP (mg/L)	0.958 (0.905–1.014)	0.044 *	0.563 (0.242–1.312)	<0.0001 *
Creatinine (µmol/L)	0.990 (0.965–1.017)	0.919	0.984 (0.950–1.020)	0.843
TcPO_2_ (mmHg)	1.134 (0.933–1.380)	0.021 *	1.104 (0.926–1.317)	0.071
ABI	5.084 (0.347–126.09)	0.199	46.635 (0.393–5533.96)	0.035
BM-MNCs (109 cells/mL)	1.560 (0.896–3.138)	0.189	1.0 (0.999–1.0)	0.035

OR, odds ratio; CI, confidential interval; CRP, C-reaction protein; ABI, ankle-brachial index; TcPO_2_, transcutaneous oxygen pressure. * The difference between six months and baseline in responders was statistically significant.

**Table 6 biomedicines-12-00922-t006:** Characteristics of patients before stem cell treatment.

	All Patients (N = 33)	ATV Group (N = 22)	Non-ATV Group (N = 11)	*p* Value
Age (years) (mean ± SD)	64.9 ± 10	64.3 ± 11	65.9 ± 9.2	0.983
Sex (males)	31 (94%)	21 (95%)	10 (91%)	1.000
Rutherford class (1–6)	5 ± 0.30	5 ± 0.21	5 ± 0.45	0.829
Body mass index (kg/m^2^)	27 ± 4.1	27.5 ± 3.8	26.9 ± 4.9	0.080
	Risk factors of limb ischaemia			
Diabetes mellitus (N, %)	11 (33%)	8 (22%)	3 (27%)	0.709
Arterial hypertension	27 (82%)	19 (86%)	8 (73%)	0.375
Hyperlipidaemia	23 (70%)	17 (77%)	6 (55%)	0.240
Smoker	10 (30%)	5 (23%)	5 (45%)	0.240
	Blood examination			
CRP (mg/L)	13.3 ± 21.3	12.8 ± 24.6	14.2 ± 14	0.341
Creatinine (µmol/L)	95.9 ± 34.6	97 ± 37.6	94 ± 29.7	0.977
	Parameters of limb ischaemia			
TcPO_2_ < 10 mmHg	19 (58%)	11 (50%)	8 (73%)	0.278
ABI	0.51 ± 0.38	0.51 ± 0.36	0.52 ± 0.47	0.951
Pain scale (0–10)	5.75 ± 1.6	5.74 ± 1.52	5.78 ± 1.86	0.951

Data are presented as mean ± standard deviation or numbers and percentages, where appropriate. CRP, C-reaction protein; TcPO_2_, transcutaneous oxygen pressure; ABI, ankle-brachial index. Statistical analysis was performed by the Mann-Whitney test and by Student’s *t*-test (depending on data distribution determined by the Shapiro-Wilk test) or Fisher’s exact test. *p* = ATV vs. non-ATV group.

**Table 7 biomedicines-12-00922-t007:** Outcomes of BMCs treatment after three and six months in all patients with or without atorvastatin treatment prior to BMCs transplant.

	ATV Group (N = 22)				Non-ATV Group (N = 11)			
	before BMCs	Three Months	Six Months	*p* Value	before BMCs	Three Months	Six Months	*p* Value
Rutherford category	4.95 ± 0.21	4.95 ± 0.60	4.55 ± 1.1	0.092	5.0 ± 0.45	5.0 ± 0.78	5.0 ± 0.93	0.655
TcPO_2_ (mmHg)	10.72 ± 13.01	14.4 ± 19.9	17.3 ± 16.5	0.015 *	7.5 ± 12.46	10.7 ± 14.9	10.2 ± 14.7	0.611
ABI	0.51 ± 0.36	0.44 ± 0.41	0.42 ± 0.34	0.532	0.52 ± 0.47	0.47 ± 0.47	0.38 ± 0.51	0.655
Pain scale (0–10)	5.74 ± 1.52	4.45 ± 2.7	3.6 ± 3.35	0.004 *	5.78 ± 1.86	5.2 ± 3.42	4.8 ± 4.07	0.202

Data are presented as mean ± standard deviation. ABI, ankle-brachial index; BMCs, bone marrow cells; tcPO_2_, transcutaneous oxygen pressure. Statistical analysis was performed by the Wilcoxon matched-pair test and by Student’s *t*-test (depending on data distribution determined by the Shapiro-Wilk test). *p* = six months vs. baseline. * The difference between six months and baseline in responders was statistically significant.

**Table 8 biomedicines-12-00922-t008:** Characteristics of transplanted BMCs.

	ATV Group (N = 22)	non-ATV Group (N = 11)	*p* Value
BM-MNCs (10^9^ cells/mL)	3.64 ± 1.53	2.58 ± 0.73	0.038
Viability of BM-MNCs (%)	99.5 ± 0.57	99.6 ± 0.67	0.537
MSCs (10^4^ cells/mL)	0.79 ± 0.52	1.47 ± 1.62	0.281

Data are presented as mean ± standard deviation. BMCs, bone marrow cells; BM-MNCs, bone marrow mononuclear cells; MSCs, mesenchymal stem cells. The Mann-Whitney test was used to perform statistical analysis. *p* = ATV vs. non-ATV group.

**Table 9 biomedicines-12-00922-t009:** Characteristics of patients before stem cell treatment.

	All Patients (N = 33)	RAS Group (N = 20)	Non-RAS Group (N = 13)	*p* Value
Age (years) (mean ± SD)	64.9 ± 10	68.65 ± 8.82	59.0 ± 9.80	0.006 *
Sex (males)	31 (94%)	19 (95%)	12 (92%)	1.000
Rutherford class (1–6)	5 ± 0.30	4.95 ± 0.22	5.0 ± 0.41	0.652
Body mass index (kg/m^2^)	27 ± 4.1	26.56 ± 2.85	28.47 ± 5.57	0.385
	Risk factors of limb ischaemia			
Diabetes mellitus (N, %)	11 (33%)	8 (40%)	3 (23%)	0.277
Arterial hypertension	27 (82%)	19 (95%)	8 (62%)	0.025 *
Hyperlipidaemia	23 (70%)	14 (70%)	9 (69%)	1.000
Smoker	10 (30%)	4 (21%)	6 (46%)	0.244
	Blood examination			
CRP (mg/L)	13.3 ± 21.3	16.09 ± 26.1	9.26 ± 10.77	0.367
Creatinine (µmol/L)	95.9 ± 34.6	100.1 ± 37.77	89.69 ± 29.8	0.258
	Parameters of limb ischaemia			
TcPO_2_ < 10 mmHg	19 (58%)	10 (50%)	9 (69%)	0.310
ABI	0.51 ± 0.38	0.59 ± 0.37	0.38 ± 0.37	0.165
Pain scale (0–10)	5.75 ± 1.6	5.56 ± 1.42	6.10 ± 1.91	0.426

Data are presented as mean ± standard deviation or numbers and percentages, where appropriate. CRP, C-reaction protein; TcPO_2_, transcutaneous oxygen pressure; ABI, ankle-brachial index. Statistical analysis was performed by the Mann-Whitney test and by Student’s *t*-test (depending on data distribution determined by the Shapiro-Wilk test) or Fisher’s exact test. *p* value = RAS vs. non-RAS group. * The difference between six months and baseline in responders was statistically significant.

**Table 10 biomedicines-12-00922-t010:** Outcomes of BMCs treatment after six months in all patients with or without RAS treatment prior to BMCs transplant.

Parameters of Limb Ischaemia	RAS Group			Non-RAS Group		
	Baseline (N = 20)	Six Months (N = 18)	*p* Value	Baseline (N = 13)	Six Months (N = 10)	*p* Value
Rutherford class (1–6)	4.95 ± 0.22	4.61 ± 0.41	0.164	5.0 ± 0.41	4.8 ± 1.03	0.276
TcPO_2_ (mmHg)	11.4 ± 13.5	15.5 ± 14.5	0.077	6.7 ± 11.4	14.3 ± 18.8	0.201
ABI	0.59 ± 0.37	0.53 ± 0.41	0.408	0.38 ± 0.37	0.38 ± 0.26	0.500
Pain scale (0–10)	5.56 ± 1.42	3.33 ± 3.25	0.005 *	6.10 ± 1.91	4.22 ± 3.83	0.139

Data are presented as mean ± standard deviation. ABI, ankle-brachial index; BMCs, bone marrow cells; TcPO_2_, transcutaneous oxygen pressure. Statistical analysis was performed using the Wilcoxon matched-pair test and Student’s *t*-test (depending on data distribution determined by the Shapiro-Wilk test). *p* value = six months vs. baseline. * The difference between six months and baseline in responders was statistically significant.

**Table 11 biomedicines-12-00922-t011:** Characteristics of transplanted BMCs.

	RAS Group (N = 18)	non-RAS Group (N = 12)	*p* Value
BM-MNCs (10^9^ cells/mL)	3.44 ± 1.55	3.05 ± 1.61	0.462
Viability of BM-MNCs (%)	99.6 ± 0.49	99.5 ± 0.75	0.859
MSCs (10^4^ cells/mL)	0.86 ± 0.60	1.23 ± 1.50	0.730

Data are presented as mean ± standard deviation. BMCs, bone marrow cells; BM-MNCs, bone marrow mononuclear cells; MSCs, mesenchymal stem cells. The Mann-Whitney test was used to perform statistical analysis. *p* value = RAS. vs. non-RAS group.

**Table 12 biomedicines-12-00922-t012:** Characteristics of patients before stem cell treatment.

	All Patients (N = 33)	ATV and RAS Group (N = 17)	Non-ATV and Non-RAS Group (N = 8)	*p* Value
Age (years) (mean ± SD)	64.9 ± 10	67.41 ± 9	62.25 ± 7.94	0.026 *
Sex (males)	31 (94%)	16 (94%)	7 (88%)	1.000
Rutherford class (1–6)	5 ± 0.30	4.94 ± 0.24	5 ± 0.54	0.817
Body mass index (kg/m^2^)	27 ± 4.1	26.37 ± 2.73	26.87 ± 5.50	0.763
	Risk factors of limb ischaemia			
Diabetes mellitus (N, %)	11 (33%)	7 (41%)	2 (25%)	0.661
Arterial hypertension	27 (82%)	16 (94%)	5 (63%)	0.081
Hyperlipidaemia	23 (70%)	13 (76%)	5 (63%)	0.640
Smoker	10 (30%)	4 (24%)	5 (63%)	0.087
	Blood examination			
CRP (mg/L)	13.3 ± 21.3	16.06 ± 27.51	13.43 ± 12.05	0.653
Creatinine (µmol/L)	95.9 ± 34.6	100.7 ± 41.31	93.06 ± 35.29	0.425
	Parameters of limb ischaemia			
TcPO_2_ < 10 mmHg	9.6 ± 12.7	8 (47%)	6 (75%)	0.234
ABI	0.51 ± 0.38	0.54 ± 0.33	0.34 ± 0.32	0.540
Pain scale (0–10)	5.75 ± 1.6	5.80 ± 1.42	6.50 ± 1.87	0.063

Data are presented as mean ± standard deviation or numbers and percentages, where appropriate. CRP, C-reaction protein; TcPO_2_, transcutaneous oxygen pressure; ABI, ankle-brachial index. Statistical analysis was performed using the Mann-Whitney test and Student’s *t*-test (depending on data distribution determined by the Shapiro-Wilk test) or Fisher’s exact test. *p* = ATV and RAS group vs. non-ATV and non-RAS group. * The difference between six months and baseline in responders was statistically significant.

**Table 13 biomedicines-12-00922-t013:** Outcomes of BMCs treatment after six months in all patients with or without ATV-RAS treatment prior to BMCs transplant.

Parameters of Limb Ischaemia	ATV and RAS Group			Non-ATV and Non-RAS Group		
	Baseline (N = 17)	Six Months (N = 15)	*p* Value	Baseline (N = 8)	Six Months (N = 7)	*p* Value
Rutherford class (1–6)	4.94 ± 0.24	4.56 ± 1.15	0.164	5.0 ± 0.54	5.0 ± 1.10	0.655
TcPO_2_ (mmHg)	11.6 ± 13.5	16.5 ± 15.0	0.033 *	6.3 ± 11.8	10.9 ± 16.4	0.344
ABI	0.54 ± 0.33	0.47 ± 0.36	0.409	0.34 ± 0.32	0.18 ± 0.32	0.650
Pain scale (0–10)	5.80 ± 1.42	3.50 ± 3.43	0.009 *	6.50 ± 1.87	5.17 ± 4.45	0.279

Data are presented as mean ± standard deviation. ABI, ankle-brachial index; BMCs, bone marrow cells; TcPO_2_, transcutaneous oxygen pressure. Statistical analysis was performed using the Wilcoxon matched-pair test and Student’s *t*-test (depending on data distribution determined by the Shapiro-Wilk test). *p* = six months vs. baseline. * The difference between six months and baseline in responders was statistically significant.

**Table 14 biomedicines-12-00922-t014:** Characteristics of transplanted BMCs.

	ATV and RAS Group (N = 17)	Non-ATV and Non-RAS Group (N = 8)	*p* Value
BM-MNCs (10^9^ cells/mL)	3.56 ± 1.60	2.60 ± 0.81	0.064
Viability of BM-MNCs (%)	99.6 ± 0.51	99.6 ± 0.75	0.642
MSCs (10^4^ cells/mL)	0.78 ± 0.56	1.53 ± 1.89	0.549

BMCs, bone marrow cells; BM-MNCs, bone marrow mononuclear cells; MSCs, mesenchymal stem cells. The Mann-Whitney test was used to perform statistical analysis. *p* = ATV and RAS vs. non-ATV and non-RAS group.

**Table 15 biomedicines-12-00922-t015:** Results of Spearman correlation analysis (R) for the investigated treatment option and baseline characteristics of patients.

		Responder	Age	CRP	RAS-Acting Agents Treatment
Responder	Spearman’s r	-	0.430	−0.442	0.351
	*p* value	-	0.012	0.011	0.045
Statin treatment	Spearman’s r	-	-	-	0.482
	*p* value	-	-	-	0.005
RAS-acting agents treatment	Spearman’s r	0.351	0.566	-	-
	*p* value	0.045	0.0006	-	-
Statin and RAS-acting agents treatment	Spearman’s r	0.618	0.454	-	1
	*p* value	0.001	0.023	-	<0.0001

## Data Availability

The data that support the findings of this study are available upon request.

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
