# Peer review of "Atorvastatin Treatment Significantly Increased the Concentration of Bone Marrow-Derived Mononuclear Cells and Transcutaneous Oxygen Pressure and Lowered the Pain Scale after Bone Marrow Cells Treatment in Patients with “No-Option” Critical Limb Ischaemia"

_biomedicines, 2024, doi:10.3390/biomedicines12040922_

Round 1

Reviewer 1 Report

Comments and Suggestions for Authors

The topic is very intriguing.

yet several data can be improved and analyzed or added in a further table.

-has thrombophilia  analyzed ?

-amtiplatelets : which type ? Aspirin ? Clopideogrel? Others? Dual

antiplatelet therapy ?

-rivaroxaban 2.5 mg : did some patients use it ? And warfarin?

-how many patients were affected by atrial fibrillation?

Author Response

Thank you for your questions. Our answers and the changes in the text note in red.

The topic is very intriguing.

yet several data can be improved and analyzed or added in a further table.

-has thrombophilia  analyzed? No. We didn't analyze the status of thrombophilia.

-antiplatelets: which type? Aspirin? Clopideogrel? Others? Dual We give in the table 1 and 3 the information about treatment with aspirin and clopidogrel. Other antiplatelet therapy the patients didn't use.

-rivaroxaban 2.5 mg : did some patients use it? And warfarin? The patients didn't use rivaroxaban or warfarin.

-how many patients were affected by atrial fibrillation? We didn't analyze this parameter – it wasn't included in the exclusion and inclusion criteria.

Reviewer 2 Report

Comments and Suggestions for Authors

In this article, Kyselovic et al reviewed thirty-three patients with advanced critical limb ischaemia (CLI) after failed or impossible revascularisation, who were treated with 40 ml of autologous BMCs by local intramuscular application. Patients with limb salvage and wound healing (n=22) were considered as responders to BMCs therapy, and patients with limb salvage and complete ischemic wound healing (n=13) were defined as super-responders. Logistic regression models were used to screen and identify the prognostic factors, and a receiver operating characteristics (ROC) curve, a linear regression and a survival curve were drawn to determine the predictive accuracy, the correlation between the candidate predictors and the risk of major amputation. Based on the univariate regression analysis, baseline C-reactive protein (CRP) and transcutaneous oxygen pressure (TcPO2) values were identified as prognostic factors of the responders, while CRP value, ankle-brachial index (ABI) and bone marrow-derived mononuclear cells (BM-MNCs) concentration were identified as prognostic factors of the super-responders. An area under the ROC curve of 0.768 indicated good discrimination for CRP ˃ 8.1 mg/L before transplantation as a predictive factor for negative clinical response. Linear regression analysis revealed a significant dependence between the levels of baseline CRP and the concentration of BM-MNCs in transplanted bone marrow. Patients taking atorvastatin before BMCs treatment (n=22) had significantly improved TcPO2 and reduced pain scale after BMCs transplant, compared to the non-atorvastatin group. Statin treatment was associated with reduced risk for major amputation. However, the difference was not statistically significant. Statin use was also associated with a significantly higher concentration of BM-MNCs in the transplanted bone marrow compared to patients without statin treatment. Patients treated with RAS-acting agents (n=20) had significantly reduced pain scale after BMCs transplant, compared to the NON-RAS group. Similar results, reduced pain scale and improved TcPO2, were achieved in patients treated with atorvastatin and RAS-acting agents (n=17) before BMCs treatment. Results of the Spearman correlation showed a significant positive correlation between CLI regression, responders, and previous therapy before BMCs transplant with RAS-acting agents alone or with atorvastatin. They concluded that CRP and TcPO2 were prognostic factors of the responders, while CRP value, ABI and BM-MNCs concentration were identified as predictive factors of the super-responders. Atorvastatin treatment was associated with a significantly increased concentration of BM-MNCs in bone marrow concentrate and higher TcPO2 and lower pain scale after BMCs treatment in CLI patients. Similarly, reduced pain scales and improved TcPO2 were achieved in patients treated with atorvastatin and RAS-acting agents before BMCs treatment. Positive correlations between responder and previous treatment before BMCs transplant with RAS-acting agents alone or with atorvastatin were significant. It is very interesting results, but I have some questions as follows.

major concerns)

1) In table 5, only univariate analysis was performed, but in order to obtain independent risk factors, multivariate analysis should also be performed.

2) It is better to perform univariate and multivariate analysis for the presence/absence of RAS and the presence/absence of ATV, rather than for the presence/absence of ATV and the presence/absence of RAS separately. Therefore, please put disparate Figs and tables together. Two important factors and outcomes are 1) presence/absence of responder and 2) presence/absence of super-responder. Please perform univariate and multivariate analyses with these two as outcomes.

Author Response

Thank you for your questions. Our answers and the changes in the text note in red.

  • In table 5, only univariate analysis was performed, but in order to obtain independent risk factors, multivariate analysis should also be performed. Multivariate analysis was feasible, even in responders and superresponders, and, as expected, did not yield any clinically relevant conclusions. For responders, only age was shown for age, BMI, creatinine, CRP, TCPO2, ABI and BM-MNCs.
  •  
  • It is better to perform univariate and multivariate analysis for the presence/absence of RAS and the presence/absence of ATV, rather than for the presence/absence of ATV and the presence/absence of RAS separately. Therefore, please put disparate Figs and tables together. Two important factors and outcomes are 1) presence/absence of responder and 2) presence/absence of super-responder. Please perform univariate and multivariate analyses with these two as outcomes. Treatment with both investigated pharmacotherapies, statin and RAS treatment showed a significant positive correlation with CLI regression, responder (R=0.618, P=0.001) and age of patients (R=0.454, P=0.023).- table 15.

Reviewer 3 Report

Comments and Suggestions for Authors

The present study researched the outcomes and potential predictive factors associated with autologous bone marrow cells therapy in patients afflicted with "no-option" critical limb ischemia. The focus was on investigating the clinical background, along with the impact of prior statin and renin-angiotensin system -acting agents pharmacotherapy on the efficacy of bone marrow cells treatment. Despite the paper has some novelty and scientific soundness, several issues must be addressed by the authors. My report is as follows:

1- Abbreviations in title are discouraged. Please spell out all acronyms in first mention in the manuscript (in abstract and in the text, separately).

2- Background data is not sufficient enough to make the study rationale clear. Please give more data in intro. Why this study was important to conduct?

3- It is not clear in methodology whether stem cell growth factors (i.e. G-CSF) were used before bone marrow aspiration.

4- Authors found that the patients who responded to the treatment were older and have lower C-reactive protein serum levels compared to the non-responders. How do the authors interpret these findings?

5- The confidence intervals of all parameters in univariate analysis in table 5 cover 1. Hence, these data should all be insignificant. Check please.

6- I am agree with the authors in term of limitations. Moreover, single center nature of the work is another limitation that should be acknowledged.

Author Response

Thank you for your questions. Our answers and the changes in the text note in red.

1- Abbreviations in title are discouraged. Please spell out all acronyms in first mention in the manuscript (in abstract and in the text, separately). We checked all abbreviations in the text, and we corrected the title of the manuscript.

2- Background data is not sufficient enough to make the study rationale clear. Please give more data in intro. Why this study was important to conduct? While previous research has proposed these theoretical frameworks, their efficacy in real-world clinical settings remains unestablished. Existing therapies for cardiovascular diseases provide a readily available platform for clinical verification of these hypotheses through a simplified clinical design.

We rewrite the aim in the background: While previous research has proposed these theoretical frameworks, their efficacy in real-world clinical settings remains unestablished. Existing therapies for cardiovascular diseases provide a readily available platform for clinically verifying these hypotheses through a simplified clinical design. This study investigates the baseline clinical characteristics and prior medication regimens by "no-option" chronic limb ischemia (CLI) patients treated with bone marrow cell therapy. We aim to identify factors associated with treatment response by comparing responders and non-responders. This analysis will contribute to developing and refining selection criteria and prognostic factors for predicting positive clinical outcomes.

3- It is not clear in methodology whether stem cell growth factors (i.e. G-CSF) were used before bone marrow aspiration. We didn't use the stem cell growth factors before the bone marrow aspiration.

4- Authors found that the patients who responded to the treatment were older and have lower C-reactive protein serum levels compared to the non-responders. How do the authors interpret these findings? We do not give relevant weight to the age value, but low CRP values point to a non-existent or clinically relevant low inflammatory process that positively affects cell treatment.

5- The confidence intervals of all parameters in univariate analysis in table 5 cover 1. Hence, these data should all be insignificant. Check please. We recount all confidence intervals; the slight differences are red in table 5.

6- I am agree with the authors in term of limitations. Moreover, single center nature of the work is another limitation that should be acknowledged. We gave this information in the limitations.

Round 2

Reviewer 1 Report

Comments and Suggestions for Authors

Conclusions need to be attenuated 

Author Response

Thank you.

We rewrite the conclusion; see please in the attachment.

Reviewer 2 Report

Comments and Suggestions for Authors

No additional comments.

Author Response

Thank you.

Reviewer 3 Report

Comments and Suggestions for Authors

Authors revised the paper accordingly. There is nothing more that need further revision. Hence, i recommend publication of the paper in the journal in its current form. 

Author Response

Thank you.